# Sensitive Detection of Trace Explosives by a Self-Assembled Monolayer Sensor

**DOI:** 10.3390/mi14122179

**Published:** 2023-11-29

**Authors:** Weitao Liu, Wajid Ali, Ye Liu, Mingliang Li, Ziwei Li

**Affiliations:** 1Hunan Institute of Optoelectronic Integration, College of Materials Science and Engineering, Hunan University, Changsha 410082, China; 2Department of Chemistry, The University of Hong Kong, Hong Kong 999077, China

**Keywords:** self-assembled monolayer, aggregation-induced emission, fluorescence, trace explosive

## Abstract

Fluorescence probe technology holds great promise in the application of trace explosive detection due to its high sensitivity, fast response speed, good selectivity, and low cost. In this work, a designed approach has been employed to prepare the TPE-PA-8 molecule, utilizing the classic aggregation-induced emission (AIE) property of 1,1,2,2-tetraphenylethene (TPE), for the development of self-assembled monolayers (SAMs) targeting the detection of trace nitroaromatic compound (NAC) explosives. The phosphoric acid acts as an anchoring unit, connecting to TPE through an alkyl chain of eight molecules, which has been found to play a crucial role in promoting the aggregation of TPE luminogens, leading to the enhanced light-emission property and sensing performance of SAMs. The SAMs assembled on Al_2_O_3_-deposited fiber film exhibit remarkable detection performances, with detection limits of 0.68 ppm, 1.68 ppm, and 2.5 ppm for trinitrotoluene, dinitrotoluene, and nitrobenzene, respectively. This work provides a candidate for the design and fabrication of flexible sensors possessing the high-performance and user-friendly detection of trace NACs.

## 1. Introduction

The escalating global threats of terrorism and the potential public safety risks posed by military explosives have instigated extensive research and innovation in explosive detection technologies [1,2]. The pressing demand for national defense security has stimulated profound investigations and advancements in explosive detection technologies to effectively address these challenges [3]. Attaining a high level of sensitivity and selectivity in explosive detection techniques holds paramount importance for counterterrorism efforts, safeguarding homeland security, and ensuring environmental protection [4]. While conventional approaches such as gas chromatography and liquid chromatography exhibit commendable sensitivity, their intricate procedures and limitations in real-time detection underscore the significance of exploring rapid explosive detection methods [5,6]. In recent years, optical materials, particularly fluorescent probes, have garnered considerable attention owing to their simplicity, portability, and swift response characteristics, thereby offering notable technical advantages in the field of explosives detection [7,8,9,10].

Traditional fluorescent probe materials often suffer from a common drawback known as aggregation-induced quenching (AIQ), which results in diminished fluorescence intensity when the materials are in concentrated solutions or solid states [11,12,13]. This inherent limitation considerably hinders the practical application of fluorescent materials in sensing technologies. In order to overcome the AIQ effect and enhance the performance of fluorescent materials for the detection of trace nitroaromatic explosives, researchers have explored the concept of aggregation-induced emission (AIE) in molecules [14,15]. Materials exhibiting the AIE phenomenon display weak fluorescence in their individual monomeric state but demonstrate significantly enhanced fluorescence upon aggregation. This intriguing phenomenon stems from the restriction of intramolecular rotation and alterations in aggregate morphology, thereby broadening the application potential of fluorescent materials [16,17,18].

Recently, scientists have shifted their attention towards AIE compounds as a potential remedy, particularly in the realm of nonaqueous phase liquid (NAPL) detection [19,20]. In order to explore pioneering methodologies, the utilization of AIE compounds, such as 1,1,2,2-tetraphenylethene (TPE), has exhibited tremendous potential in NAPL detection. Nevertheless, prior investigations have predominantly centered around liquid-phase systems, thereby impeding their applicability in practical solid-state devices. The pursuit of swift response times and exceptional sensitivity continues to pose a formidable challenge, which needs the existence of high-concentration SAMs in a limited area or volume [21,22,23].

In our previous research, TPE was constructed as a sequence of functional molecules, TPE-PA-n (n = 3–11), wherein each molecule possessed a distinct alkyl chain length [24,25]. These molecules have been employed as highly sensitive solid-state fluorescent probes for trace explosives detection, which work based on the properties of AIE. Experimental analysis has revealed that shorter alkyl chains exhibit limited flexibility and longer alkyl chains restrict the coupling between molecules. Among the range of alkyl chains investigated (n = 3–11), TPE-PA-8 with the advantages of strong fluorescence intensity and high quenching efficiency, has been demonstrated as an ideal candidate for practical SAM sensor preparation. However, in planar thin-film TPE structure, the optical cross-section of light matter interaction is still limited, compared to TPE constructed on nanophotonic structures. To further enhance the sensitivity detection ability of TPE, designing and fabricating materials on nanophotonic structures provides a promising technological route.

In this research, we present a novel approach for the preparation of a flexible sensor utilizing the AIE property of TPE-PA-8 for the detection of NAC explosives. The designed molecule consists of TPE luminogens connected to a phosphoric acid anchoring unit through an alkyl chain of eight molecules, which plays a crucial role in promoting the aggregation of TPE luminogens. These designed SAMs cooperating with fiber structures gain enhanced light-emission properties and sensing performances. The ability of fluorescence detection detected by spectral analyses reveals that the SAM sensor exhibits remarkable performances with detection limits of 0.68 ppm, 1.68 ppm, and 2.5 ppm for trinitrotoluene, dinitrotoluene, and nitrobenzene, respectively. Our work provides a promising candidate for the design and fabrication of flexible sensors that possess high-performance, cost-effectiveness, and user-friendliness in detecting trace NACs.

## 2. Materials and Method

### 2.1. Materials

All reagents and chemicals were sourced from Sigma Aldrich (Saint Louis, MO, USA) and J&K (Beijing, China) and unless specifically stated, did not require additional purification before use. The reactions were performed using the conventional Schlenk technique within an inert argon atmosphere, utilizing anhydrous solvents. The heavily doped n-type silicon wafers underwent an initial cleaning in a Piranha solution (sulfuric acid to hydrogen peroxide volume ratio of 70:30), followed by heating at 110 °C for two hours, a thorough rinse with deionized water, and an RCA clean (a mixture of deionized water, ammonium hydroxide, and hydrogen peroxide in a volume ratio of 5:1:1), before being dried with nitrogen gas for immediate use. Fibers in PMMA solution were spin-coated on the PET substrate forming the complex interleaved network. A 10 nm thick film of Al_2_O_3_ was deposited using the ALD system at 200 °C after 110 cycles. Before the growth of SAMs, flame treatment was employed to strip organic adhesives and other contaminants from the samples. Then, Al_2_O_3_-coated wafers/fiber samples were immersed at room temperature in a 0.1 mM TPE-PA-8 solution in tetrahydrofuran (THF) for 24 h under an argon atmosphere to form the SAMs. After incubation, substrates were ultrasonically treated in THF and ethanol, each for 5 min in three cycles, to remove any surface-adsorbed species. They were then dried with nitrogen gas and stored in a glovebox for subsequent use. 

For sensing tests of NACs, the experiments were conducted by hanging designed SAM-coated fiber films in a closed container, where NACs were dissolved in methanol solutions and located at the bottom of the container (See Appendix A). Different concentrations of NAC solutions were first taken in an enclosed 20 mL glass bottle, with 4 microliters of solution in each case. Once the solutions were fully evaporated and stabilized, SAM-coated fiber films were completely immersed in them for a duration of 1 min (Appendix A). High-explosive substances such as TNT (2,4,6-trinitrotoluene), DNT (2,4-dinitrotoluene), and NB (nitrobenzene) were selected as solutes and dissolved in methanol within a concentration of 1 mg/mL. After the TPE-PA-8 SAM-coated fiber film was positioned in the bottle using a bracket for five minutes, the fiber film was then picked out for fluorescence spectrum detection.

### 2.2. Characterization and Method

X–ray photoelectron spectroscopy: XPS data were acquired using an Axis Ultra Imaging XPS (Manchester, UK) equipped with a 300 W AlKα radiation source, operating at a base pressure of approximately 3 × 10^−9^ mbar, with the binding energy scale being calibrated against the C1s peak at 284.8 eV.

Scanning electron microscope: The morphology of nanostructures was characterized by the field-emission SEM (Nova NanoSEM 230, FEI, Los Angeles, CA, USA), with a scanning voltage setting at 1.0 kV. 

High–resolution transmission electron microscopy: TEM images were performed via FEI Tecnai GX F20. HRTEM image was acquired using an image and probe Cs-corrected Thermofisher Themis Z operated at 300 kV. 

X–ray reflectivity: XRR measurements were conducted on a Bruker D8-Advance diffractometer with λ = 0.154 nm. The qz vector was perpendicular to the sample surface. Reflectivity, as a function of qz (where qz=4π/(λsinθ), was normalized to the incident beam intensity. The R(qz) profiles, highly sensitive to the electron density along the surface normal, allowed for independent ascertainment of SAM thickness, density, and roughness. Motofit was employed to fit the XRR data, positing a two-layer composition of an Al_2_O_3_ base and TPE-PA-8 SAMs.

Contact Angle: Contact angle measurements were performed on a Biolin THETA optical tensiometer (Danderyd, Sweden).

Atomic force microscope: AFM measurements of the film morphology were conducted using the ScanAsyst model (Bruker Dimension Icon with Nanoscope V controller (Saarbrucken, Germany) in ambient conditions. The data were processed using Nano Scope Analysis software version 1.8. 

UV–Visible absorption: The UV spectra for both solutions (with acetonitrile as the solvent) and films (on quartz substrates) were determined using the WITec Alpha300 optical system and the Horriba iHR 550 spectrophotometer (Kyoto, Japan).

Fluorescence spectrometer: Fluorescence measurements were carried out at room temperature using a time-correlated single-photon counting (TCSPC) spectrometer, employing a front-face approach. The steady-state spectroscopy utilized a 450 W xenon lamp as the light source, and lifetime testing was conducted using a PELED260 (Edinburgh, UK). 

### 2.3. Calculation

The quenching efficiency was calculated by Equation (1).
(1)$$\gamma =\left(1-\frac{F}{F_{0}}\right)\times 100\%$$

In this equation, γ represents the quenching efficiency, $F_{0}$ is the fluorescence intensity at time t in the absence of a quencher, and $F_{0}$ is the fluorescence intensity of the film at time t. If the fluorescence quenching efficiency is greater than the upper limit for methanol detection, the detection limits for NACs can be inferred using Equation (1).
(2)$$LOD=\frac{N_{nac}}{N_{nac}+N_{meth}+N_{air}}=\frac{\frac{V_{nac}C}{M_{nac}}}{\frac{\rho_{meth}V_{meth}}{M_{meth}}+\frac{V_{nac}C}{M_{nac}}+\frac{\rho_{air}V_{air}}{M_{air}}}$$
where *LOD* is the detection limit. $N_{nac}$, $N_{meth},$ and $N_{air}$ are the NACs, air, and methanol molecule number, respectively. $V_{nac}$ is the volume of the NACs solution. $\rho_{meth}$, $V_{meth},$ and $M_{meth}$ are the density, volume, and molar mass of methanol, respectively. In the situation of small concentration, $\rho_{meth}$ is regarded as 0.7918 g mL^−1^. $V_{meth}$ and $V_{nac}$ are 4 μL. Simplified detection limit calculations can be derived through the molecular counts of NACs, air, and methanol in the instrument (20 mL) along with the densities, volumes, and molar masses of methanol and air, where they are 1.29 kg m^−3^, 20 mL, and 29 g mol^−1^, respectively.
(3)$$D=\frac{4046{NAC}_{C}}{{NAC}_{M\;}}$$

${NAC}_{C}$ and ${NAC}_{M}$ are the concentration and the molar mass of NACs.

## 3. Results and Discussion

Figure 1a shows the application scenario of a flexible SAM sensor on a finger for the trace explosive detection. The molecule has been specifically engineered to serve as a highly sensitive solid-state fluorescent probe with ordered superstructures and the inset shows the structure of the designed SAMs. Usually, the classic TPE is chosen as the chromophore unit, while the phosphoric acid is selected as the anchor group, and they are connected through an alkyl chain. To fabricate the device, conventional spin-coating and atomic layer deposition (ALD) have been employed to prepare the Al_2_O_3_-coated fiber layer on the substrate. First, fibers in a PMMA solution were spin-coated on the PET substrate forming the complex interleaved network. After depositing the Al_2_O_3_ adhesive layer using ALD and activating surface bonds through oxygen plasma treatment (Figure 1c, steps 1–2), the substrates were then immersed in a solution containing the functional molecules, forming SAMs (Figure 1c, step 3). The formation of stable covalent bonds between phosphoric acid and the Al_2_O_3_ layer on the surface of the flexible substrate guarantees the robustness and durability of the device. The anchoring of phosphoric acid to the aluminum oxide layer provides the necessary flexibility for self-assembly, leading to the aggregation of the TPE molecules, as shown in Figure 1b. Finally, TPE-PA-8 SAMs undergo fluorescence quenching in the presence of NACs, showing strong fluorescence intensity and high sensitivity (Figure 1c, step 4).

Notably, during the incubation process, a substantial change in contact angle was observed. Within the first 5 min of incubation, there is a significant increase in the surface contact angle, shifting from initial values of 67.04° and 67.54° to 76.29° and 76.13°, respectively (Figure 2a,b). After 24 h, incubation increases the hydrophobic characteristics of the SAM-coated substrate with maximum contact angle values reaching to 90.52° and 90.37°, as depicted in Figure 2c. Figure 2d shows the scanning electron microscope (SEM) image of the test substrate. Scale bar is 4 µm. This complex structure enhances the specific surface area, which is advantageous for the formation and adsorption of the SAMs. Figure 2e shows the observation of smoothness of the SAM-coated alumina fibers. Scale bar is 2 µm. Figure 2f presents the cross-sectional view of the interface between fiber and SAM layer, detected from a high-resolution transmission electron microscope (HRTEM). Scale bar is 5 nm. Also, from the image of atomic force microscopy (AFM), the surface of SAM/fiber remains remarkably flat, exhibiting an average roughness of only 0.252 nm. This indicates that the SAM surface maintains a high level of smoothness and uniformity, contributing to the overall quality and consistency of the device.

X-ray photoelectron spectroscopy (XPS) measurements have been performed to verify the bonding between the functional molecules and Al_2_O_3_. These results from XPS measurements indicate the presence of the P 2p peak even after subjecting the samples to multiple cleaning cycles with organic solvents (Figure 2h). The persistence of the P 2p peak, as depicted in Figure 2f, suggests that there is a strong interaction or adsorption of phosphorus-containing species on the sample surface, which remains resistant to the organic solvent cleaning process. This observation is consistent across the multiple cleaning attempts with organic solvents. Additionally, the O 1s peak analysis revealed distinct peaks at 532.5, 532.3, and 532.1 eV, corresponding to Al-O-Al, Al-O-P (or P=O), and C-O-C bonds, respectively (Figure 2i). These findings provide further confirmation of the covalent bonding between the phosphoric acid and the substrate, establishing the successful preparation of the SAMs.

To verify the dense and uniform thickness of TPE-PA-8 SAMs, low-angle X-ray reflection (XRR) was measured, and the corresponding results are presented in Figure 3a. The XRR analysis enables the determination of the scattering length density (SLD) fitting, providing valuable insights into the structural composition of the sensor, as depicted in Figure 3b. Based on the SLD fitting, the sensor structure was found to own two layers, including an Al_2_O_3_ layer with a thickness of 10 nm, followed by the TPE-PA-8 SAM layer with a thickness of 1.9 nm. It is worth noting that, for a clearer display of the SAMs thickness, the thickness of the Al_2_O_3_ layer is not fully depicted in Figure 3b. These measurements consist of the maximum height difference observed through AFM measurements. The XRR analysis confirmed the dense and uniform nature of the TPE-PA-8 SAMs on the substrate. Furthermore, the successful bonding of TPE-PA-8 SAMs to the Al_2_O_3_-coated fibers was observed, akin to their bonding on the Al_2_O_3_-coated silicon wafer.

In order to detect the NACs using the as-prepared SAM sensor, the device is placed in a saturated nitrobenzene atmosphere for complete fluorescence quenching, as shown in Figure 3c. This spectral graph displays the fluorescence spectra of both the unquenched (original, blue line) and fully quenched states (total quenching, red line). The fluorescence intensity of SAM sensor after quenching decreased dramatically. Figure 3d shows the time-resolved fluorescence-quenching processes on both hard wafer and flexible fiber substrate, where the quenching efficiency approaches 100% after nearly 20 min. It was also found that both the wafer and test substrate responded rapidly, with quenching efficiencies of 94.3% and 93.5% in 8 min, respectively. Interestingly, the fluorescence lifetime basically remains constant throughout the quenching process as reported before [24], indicating that the quenching process of NACs is a static quenching process, whereby a fluorochrome in the ground state can form a nonemissive bound complex with the quencher. Under the light excitation, TPE-PA-8 exhibits high-efficiency emissions through radiative decay. However, when the sensor is complexed with nonaromatic compounds, a quenching of fluorescence occurs due to photo-induced electron transfer (PET). The electrons preferentially transfer from TPE-PA-8 to the NACs, facilitated by the lower energy levels of the NACs’ lowest unoccupied molecular orbitals as discussed in previous works [21]. Therefore, the presence of NACs creates a nonradiative decay channel, reducing the luminescence efficiency while having minimal effect on the luminescence lifetimes of TPE-PA-8.

To test the sensor performance, various experiments of sensors exposed to NACs were carried out. The reversibility of the device responsivity was evaluated through the measurement of fluorescence intensity. Prior to immersing the device in a saturated nitrobenzene atmosphere for 5 min, fluorescence tests were then carried out. Subsequently, the device was immersed in ethanol for 5 min to facilitate recovery, followed by another round of fluorescence tests after drying. The reversibility of the sensor device was assessed based on 20 tests, as illustrated in Figure 4a. It was observed that the device exhibited commendable reversibility, with a quenching efficiency over 92% and a recovery efficiency over 88%, indicating its capability to effectively respond to the presence and removal of the target analyte. The selectivity of the SAM probe was investigated to determine its ability to differentiate the target NACs from common reagents such as esters, ethers, acids, and ketones. Figure 4b presents the results of sensing selectivity, demonstrating that the SAM probe exhibits an optimal response to nitrobenzene (representative of NACs). This outcome demonstrates its promise in detecting NACs in complex environments, demonstrating the diversity of responses to various chemicals.

For long-term efficient operation, device stability is crucial. Thus, the fluorescence spectrometer analysis was carried out through 10 consecutive scans. These results, depicted in Figure 4c, demonstrate that the fluorescence intensity can be maintained at a level above 96.6% throughout the 10 cycles, affirming the excellent stability of the sensor device. Furthermore, sensitivity characterization is performed using TNT, DNB, and NB, as depicted in Figure 4d. These compounds possess different levels of electron deficiency due to varying numbers of nitro groups. As the most electron-deficient system, TNT exhibited the highest ability to attract electrons, resulting in significant fluorescence quenching of the SAMs. Consequently, the detection abilities and limits vary for TNT, DNB, and NB. Quenching efficiency exceeding 5% is effective, surpassing the upper limit of solvent methanol-quenching fluctuation. As a result, the detection limits for TNT, DNB, and NB were determined as 0.68 ppm, 1.68 ppm, and 2.5 ppm, respectively. The sensing performances of related materials reported in the recent literature are summarized in Table 1, and our SAM sensor devices have been proved to have good stability, reversibility, selectivity, and sensitivity. Although TPE-PA-8 does not exhibit the highest sensitivity to explosives compared with other materials, the data presented in this study demonstrate its considerable future potential due to the high selectivity exhibited in complex environments, along with its repeatability and inherent flexibility as a wearable sensor [26,27].

## 4. Conclusions

In this study, TPE-PA-8 has been successfully designed as a SAM sensor for the detection of trace NAC explosives. The SAMs are constructed by connecting a classic AIE molecule, TPE, and phosphoric acid, which serve as the chromophore and anchor, respectively. The stability of the device is ensured through the covalent bonding of SAMs to the Al_2_O_3_-coated fiber substrate, while any unbonded molecules are subsequently removed, leaving behind a monolayer film. The synthesized SAMs exhibit strong fluorescence intensity, sensitive quenching ability and excellent performance in reversibility, selectivity, and stability. It promises to be a suitable candidate for practical SAM sensor preparation. Overall, this work highlights significant potential for the development and application of flexible, wearable, and transparent devices for NAC explosive detection, providing an experimental foundation for future sensor advancements.

## Figures and Tables

**Figure 1 micromachines-14-02179-f001:**
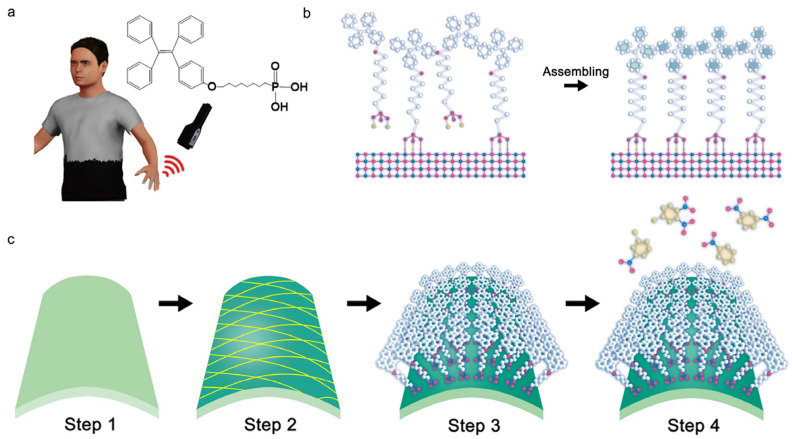
(**a**) The schematic of a flexible SAM sensor on a finger for trace explosives. Inset is the molecular structure of TPE-PA-8. (**b**) Formation and assembling of TPE-PA-8 SAMs on the Al_2_O_3_ adhesive layer. (**c**) Device fabrication process. Step 1: substrate preparation; step 2: fiber-film spin-coating and Al_2_O_3_ deposition; step 3: formation of SAM-coated fiber sensor by solution incubation; step 4: sensing and fluorescence quenching by NACs.

**Figure 2 micromachines-14-02179-f002:**
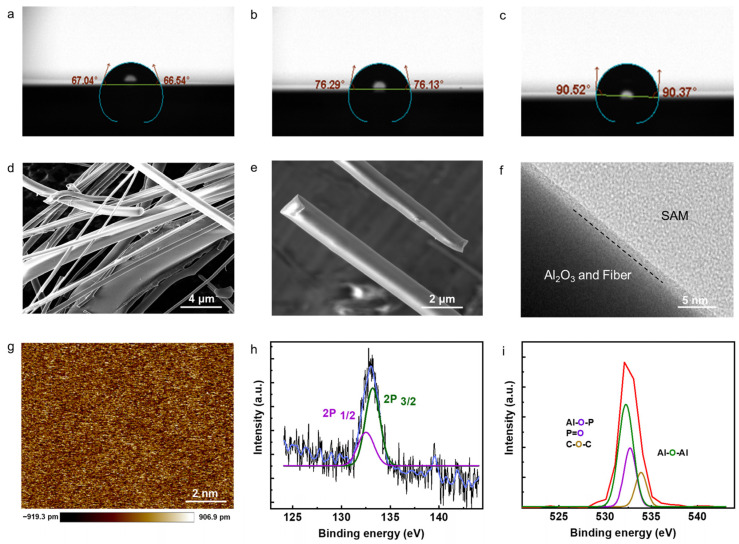
The contact angle observation of (**a**) bare substrate with Al_2_O_3_ layer and SAM-coated substrate at the time of preparation (**b**) and after 24 h aging (**c**). SEM images of the Al_2_O_3_-coated fiber film at low (**d**) and high (**e**) resolution. (**f**) HRTEM image of the interface between Al_2_O_3_/fiber and SAM layers. (**g**) AFM image of SAMs layer. (**h**) High-resolution XPS of P 2p and (**i**) O 1s.

**Figure 3 micromachines-14-02179-f003:**
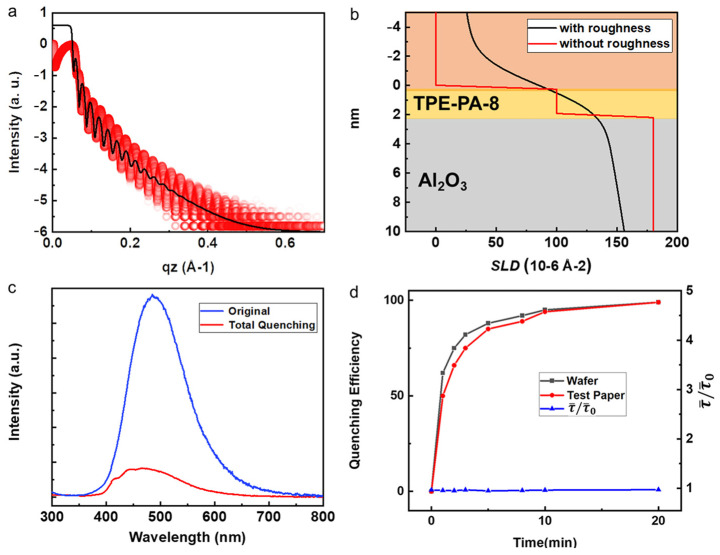
(**a**) XRR plots of the SAM sensor (red dots) and the corresponding fitting curve (black curve). (**b**) Scattering length density (SLD) profile with (black curve) and without roughness (red curve). (**c**) The fluorescence spectra detected before and after PL quenching. (**d**) Time-dependent PL quenching efficiency and lifetimes of SAMs on hard wafer and flexible fiber substrate.

**Figure 4 micromachines-14-02179-f004:**
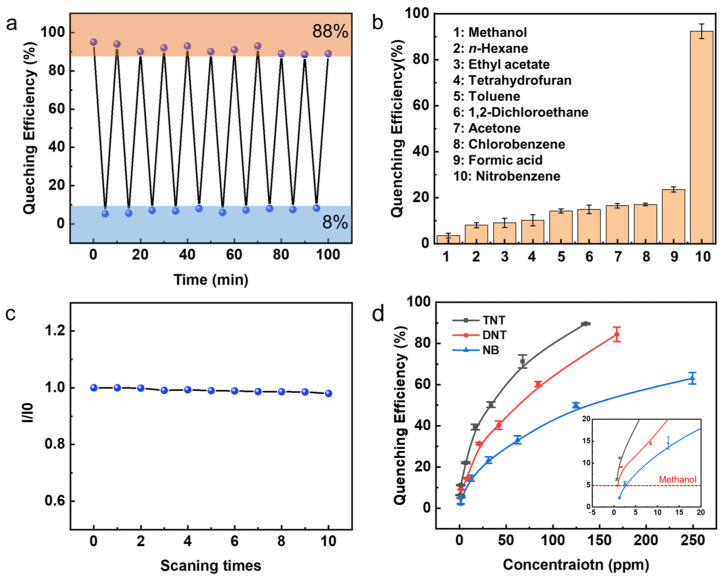
(**a**) TPE-PA-8 SAMs reversibility tests performed 20 times. (**b**) Selectivity tests for the TPE-PA-8 SAMs over 10 different materials. (**c**) The peak fluorescent efficiency (I/I_0_) stability over scanning times. (**d**) Concentration-dependent quenching of TPE-PA-8 SAMs for the NACs. Inset shows the enlarged view of quenching efficiencies at low concentrations.

**Table 1 micromachines-14-02179-t001:** Sensing performances of related materials.

Sensing Materials	NAC Explosives				
	NB	TNT	DNB	DNT	Ref.
Pyrimidine scaffold with a pyrene-donative fragment	6 ppb	5 ppb	-	-	[26]
Sol-Gel Materials	-	5 ppb	-	-	[27]
Oligomer P1	-	698 ppb	-	-	[28]
3,3′-{[1,4-phenylenebis-(methylene)] bis(oxy)} dibenzoic acid (H2L)	3.53 ppm	-	3.06 ppm	1.56 ppm	[29]
1-pyrene-based derivatives	-	3.11 ppm	-	1.82 ppm	[30]
Ultrathin PTPE	4.1 ppm	0.07 ppm	0.35 ppm	-	[21]
TPE-PA-8	2.5 ppm	0.68 ppm	1.68 ppm	-	This work

## Data Availability

The data presented in this study are available in the article.

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
