# Peer review of "Sensitive Detection of Trace Explosives by a Self-Assembled Monolayer Sensor"

_micromachines, 2023, doi:10.3390/mi14122179_

Round 1
Reviewer 1 Report
Comments and Suggestions for Authors
The paper entitled: Sensitive Detection of Trace Explosives by a Self-Assembled Monolayer Sensor, by: W. Liu, W. Ali, Y. Liu, M. Li, and Z. Li, deals with the preparation of nitroaromatics sensor by modifying the surface of aluminum oxide with an aggregation induced fluorescent monolayer, in turn supported over polymethylmethacrylate electrospun fibers. The preparation is well described and the characterization and initial experiments are convincing. An experimental supporting information is required in order to understand all preparative procedures and the performance of sensing experiments. Some of the figures are confusing. For example, Figure 3c looks as a common fluorescence spectrum, intensity versus wavelength, but instead it is explained that it is composed of measurements of quenching, this has to be clarified. Initial sensing experiments are performed in the gas phase with nitrobenzene, but the experiments with solvents are not explained, are they performed in the gas phase? In which amounts? Then, experiments are performed with other nitro aromatic compounds but this time in solution, the comparison between the previous and new experiments is not fully understandable. This fact should be clarified. A comparison with other nitroaromatics, such as picric acid, should be conducted. The method to obtain the detection limits from such experiments is not explained and the corresponding titration plots did not appear in the manuscript. This fact should be completed. In summary, the paper may be publishable as a new material for explosive detection, albeit the claimed detection limits are not very low, but the experimental part should be completed before next evaluation of the paper.
Comments on the Quality of English LanguageEnglish is fine.
Reviewer 2 Report
Comments and Suggestions for Authors
The article by Li et al. is certainly very relevant in connection with the current global situation. However, in this form it cannot be published in the journal Micromachines. There are a number of serious shortcomings that must be corrected before publication of the article becomes possible.
1. The authors should provide an explanation of why this particular fluorophore TPE-PA-8 was chosen. Is this compound known or did the authors synthesize it themselves for the first time? If the compound is known, it is necessary to provide a reference to its synthesis and a description of the photophysical and sensory properties for the detection of nitroaromatic compounds in solutions.
It is also necessary to confirm that compound TPE-PA-8 has the properties of aggregation-induced fluorescence, which the authors write a lot about in the introduction.
2. In the “Materials and Methods” section, the authors must provide a detailed description of all the devices on which the studies were performed. In addition, it is necessary to describe in detail all experimental procedures and methods for calculating detection limits of the detected analytes.
3. On page 5, the authors claim that there is a static mechanism of fluorescence quenching. It is necessary to give the corresponding graphs, and also describe how this was studied experimentally.
4. In conclusion, the authors need to provide a comparative table that would summarize the recent literature and techniques able to detect nitroaromatic explosives in the gas phase. In particular, it is necessary to add literary references to this table regarding the use of prototype sensors and devices for detecting nitrobenzene, trinitrotoluene and 2,4-dinitrotoluene: https://doi.org/10.1007/s13738-017-1278-7, https://doi.org/10.3390/s19183909, https://doi.org/10.3390/molecules27206957, https://doi.org/10.3390/nano12081278, etc.
Comments on the Quality of English LanguageMinor editing of English language required.
Round 2
Reviewer 1 Report
Comments and Suggestions for Authors
The paper entitled: Sensitive Detection of Trace Explosives by a Self-Assembled Monolayer Sensor, by: W. Liu, W. Ali, Y. Liu, M. Li, and Z. Li, deals with the preparation of nitroaromatics sensor by modifying the surface of aluminum oxide with an aggregation induced fluorescent monolayer, in turn supported over polymethylmethacrylate electrospun fibers. The preparation is well described and the characterization and initial experiments are convincing. The authors have now added some experimental details and a short supporting information useful to understand all preparative procedures and the performance of sensing experiments. A comparison with other sensing materials has been added. In summary, the paper may be publishable as a new material for explosive detection, albeit the claimed detection limits are not very low, because of the novelty of the material.
Comments on the Quality of English LanguageEnglish is fine, some minor correction is needed.
Reviewer 2 Report
Comments and Suggestions for Authors
Authors have carefully checked and modified this manuscript. Now it can be accepted for publication in this journal without further revision.
Comments on the Quality of English LanguageMinor editing of English language required.